# Muskrats are greater carriers of pathogenic *Leptospira* than coypus in ecosystems with temperate climates

**Florence Ayral[1]☉\*, Angeli Kodjo[1], Gérald Guédon[2], Franck Boué[3], Céline Richomme[3]☉**

**1** Université de Lyon, INRAE, VetAgro Sup, UsC 1233 RS2GP, Marcy-l'Etoile, France, **2** FNLON (National federation for pest control), Angers, France, **3** Nancy Laboratory for Rabies and Wildlife, ANSES, Malzéville, France

☉ These authors contributed equally to this work.
\* florence.ayral@vetagro-sup.fr

**Data Availability Statement:** All relevant data are within the manuscript and its Supporting Information files.

## Abstract

Knowledge on the possible sources of human leptospirosis, other than rats, is currently lacking. To assess the distribution pattern of exposure and infection by *Leptospira* serogroups in the two main semi-aquatic rodents of Western France, coypus (*Myocastor coypus*) and muskrats (*Ondatra zibethicus*), results of micro-agglutination testing and renal tissue PCR were used. In coypus, the apparent prevalence was 11% (n = 524, $CI_{95\%}$ = [9% - 14%]), seroprevalence was 42% (n = 590, $CI_{95\%}$ = [38% - 46%]), and the predominant serogroup was Australis (84%). In muskrats, the apparent prevalence was 33% (n = 274, $CI_{95\%}$ = [27% - 39%]), seroprevalence was 57% (n = 305, $CI_{95\%}$ = [52% - 63%]), and the predominant serogroup was Grippotyphosa (47%). Muskrats should therefore be considered an important source of Grippotyphosa infection in humans and domestic animals exposed in this part of France.

## Introduction

*Leptospira* spp. can infect many domestic and wild mammals that may shed the bacteria in their urine. Humans may acquire potentially fatal leptospirosis through direct contact with the urine of infected animals or indirectly through interaction with a urine-contaminated environment [1]. The pathogenic agents of leptospirosis are bacteria from the genus *Leptospira*. Specifically, subclade 1, historically considered to cover pathogenic species, includes 17 species, among which *L. interrogans*, *L. kirschneri* and *L. borgpetersenii* [2]. According to the serological classification, more than 250 pathogenic serovars are now recognised and clustered into 24 antigenically related serogroups [3].

According to data from the French National Reference Centre for Leptospirosis, in charge of national leptospirosis surveillance, mainland France has a higher incidence (0.5 to 1 case per 100 000 inhabitants) compared to other industrialized countries with a similar temperate climate [4]. An increase in the number of confirmed cases in mainland France has been observed in recent years, with an incidence of 1 per 100 000 inhabitants in 2014 and 2015, an

**Funding:** Sampling and analysis were mainly funded by the French Ministry of Agriculture in the framework of the GEDUVER project run by French national federation for pest control (Fédération nationale de lutte contre les organismes nuisibles, FNLON). The funders had no role in study design, data collection and analysis, decision to publish, or preparation of the manuscript.

**Competing interests:** The authors have declared that no competing interests exist.

unprecedented figure in the country since the beginning of leptospirosis surveillance in 1920 [4,5].

Leptospirosis outbreaks have previously been associated with recreational activities (canoeing, kayaking, water rafting, triathlon, and swimming) that bring people into close contact with water contaminated with pathogenic leptospires [6,7]. The climatic conditions in intertropical zones promote outbreak events [8,9]. In temperate zones such as Western Europe, leptospirosis outbreaks were rare in previous decades and were generally limited in their extent [10]. However, the increasing development of aquatic recreational activities in France, and the corresponding increasing number of people exposed to fresh water, could lead to higher leptospirosis incidence in the coming years in France. Recent studies have reported larger-scale outbreaks related to water exposure in France and neighbouring countries [11,12].

Although *Leptospira* can be maintained in aquatic environments for weeks [13], the main source of the bacterium is a wide range of domestic and wild mammals carrying specific *Leptospira* serogroups. The wild rat (*Rattus* spp.) is well documented as being the host of the Icterohaemorrhagiae serogroup worldwide, including in France [14], but little is known about the role of other wildlife species in *Leptospira* carriage, and their relative importance in human infections. Coypus, also known as nutrias (*Myocastor Coypus*), and muskrats (*Ondatra zibethicus*) are semi-aquatic rodents and significant carriers of pathogenic *Leptospira* in Europe. Human aquatic activities could lead to close contact with their habitat, with resulting public health issues [15,16]. In France, previous studies on coypus have reported prevalence between 5% and 12%, and the predominance of the Icterohaemorrhagiae serogroup [15,17,18]. However, data on muskrats are limited even though they share the same habitat and may be a source of water contamination.

To assess the relative importance of coypus and muskrats in water contamination, [1] the prevalence of *Leptospira* renal carriage, as well as *Leptospira* exposure were estimated in semi-aquatic rodents trapped in Western France, and [2] the distribution of *Leptospira* serogroups that these species have been exposed to was described.

## Materials and methods

All samples were collected from rodents legally killed for population control; therefore, this study did not involve deliberate additional killing of animals and no ethical approval was considered necessary. All procedures for population control complied with the ethical standards of the relevant national and European regulations on the care and use of animals (French authority Decision 2007/04/06 and Directive 2010/63/EC).

From September 2010 to May 2011, coypus and muskrats were trapped in wetland areas (marsh/pond or river), in each of the 12 departments (i.e., administrative units) of the Brittany, Normandy and Pays-de-la-Loire regions in the Western part of mainland France. Trapping was carried out at five randomly selected sites per department, i.e., 60 sampling sites in total. Trapping was implemented by duly authorized technicians from the departmental federation for pest control (*Fédération départementale des groupements de défense des organisames nuisibles*, FDGDON). The location of the trapped animals was defined as the GPS coordinates of the trapping area centroid. In each area, 20 traps were distributed within a 1 km transect. Traps were set for 3 to 5 days and verified daily. The captured individuals were immediately euthanized and a blood sample was collected by cardiac puncture. Subsequently, the rodents were necropsied for collection of kidneys, and immediately stored at −20˚C until further analysis within the following nine months. *Leptospira* colonization of the kidney was assessed via a pathogen-specific *Leptospira* TaqMan real-time polymerase chain reaction (PCR) kit (TaqVet PathoLept kit, LSI, France) used at the Laboratoire des Leptospires (Marcy-L'Etoile, France).

Specimens with a cycle threshold of less than 40 cycles were considered positive samples. *Leptospira* exposure was assessed using a micro-agglutination test (MAT) as the standard serological test. The MAT was performed using a panel of antigens representing both ubiquitous serovars and locally prevalent serovars, with log2 dilution series between 1:100 and 1:6400. The following *Leptospira* serogroups, with related serovars in parentheses, were screened for in both species: Icterohaemorrhagiae (Icterohaemorrhagiae, Copenhageni), Australis (Munchen, Australis, Bratislava), Autumnalis (Autumnalis, Bim), Ballum (Castelonis), Bataviae (Bataviae), Canicola (Canicola), Cynopteri (Cynopteri), Grippotyphosa (Grippotyphosa, Vanderhoedoni), Hebdomadis (Hebdomadis), Panama (Manama, Mangus), Pomona (Pomona, Mozdok), Pyrogenes (Pyrogenes), Sejroe (Sejroe, Saxkoebing, Hardjo, Wolffi) and Tarassovi (Tarassovi).

As antibodies may persist for prolonged periods after infection, no consensus is reported on the titer cut-off required to define an infected individual. However, a titer ≥1:100 with seroreactivity directed against at least one serogroup is considered to indicate previous or recent exposure of the individual. The presumptive serogroup responsible for seroreactivity is then defined based on the maximum antibody titer directed against one serogroup, as suggested by Chappel *et al.*, (2006). Cross-reactivity between serogroups frequently occurs in MAT and results from a lack of specificity, especially from predominant non-specific immunoglobulin M (IgM) antibodies at the onset of infection [19]. In these cases, MAT results involve the maximum antibody titers directed against two or more serogroups, thus preventing determination of the infecting serogroup. MAT results, including maximum antibody titers directed against two serogroups (i.e., "mixed" results), are still informative by indicating one or the other as potentially circulating. In contrast, taking into account more than two possible circulating serogroups is speculative and uninformative (i.e., "unknown" results).

Apparent prevalence, seroprevalence, and 95% confidence intervals were calculated using exact binomial tests. To assess potential variation in *Leptospira* serogroup distribution, the study area was divided into three administrative regions: Brittany, Normandy and Pays de la Loire. Data were visualized in ArcGIS version 9.3 (ESRI, Redland, CA, USA) with the background map from IGN GEOFLA®.

## Results

Among the 590 trapped coypus, *Leptospira* detection using PCR was able to be performed on 524 individuals with 59 positives, resulting in an apparent prevalence of infection of 11%, $CI_{95\%} = [9\% - 14\%]$.

The MAT-positive results on 247 coypus (42%, n = 590) showed broad exposure to *Leptospira* with MAT titer results ranging from 1:100 to 1:6400 (median: 1:3200). The predominant serogroups were Australis (84%) (Fig 1). In total, seroreactions including two serogroups were observed in 17 coypus, with combinations of Australis and Icterohaemorrhagiae (n = 16) and of Australis and Bataviae (n = 1). Among the infected coypus (PCR-positive) with a seroreaction (n = 32), the predominant serogroup remained Australis (90%, mixed results not included).

Among the 305 trapped muskrats, *Leptospira* detection using PCR was able to be performed on 274 individuals with 90 positives, resulting in an apparent prevalence of 33%, $CI_{95\%} = [27\% - 39\%]$.

The MAT-positive results on 175 muskrats (57%, n = 305) also showed broad exposure to *Leptospira* with MAT titer results ranging from 1:100 to 1:6400 (median: 1:1600). The predominant serogroups were Grippotyphosa (47%), Australis (22%) and Sejroe (14%) (Fig 1). In total, seroreactions including two serogroups were observed in 12 muskrats, with mainly

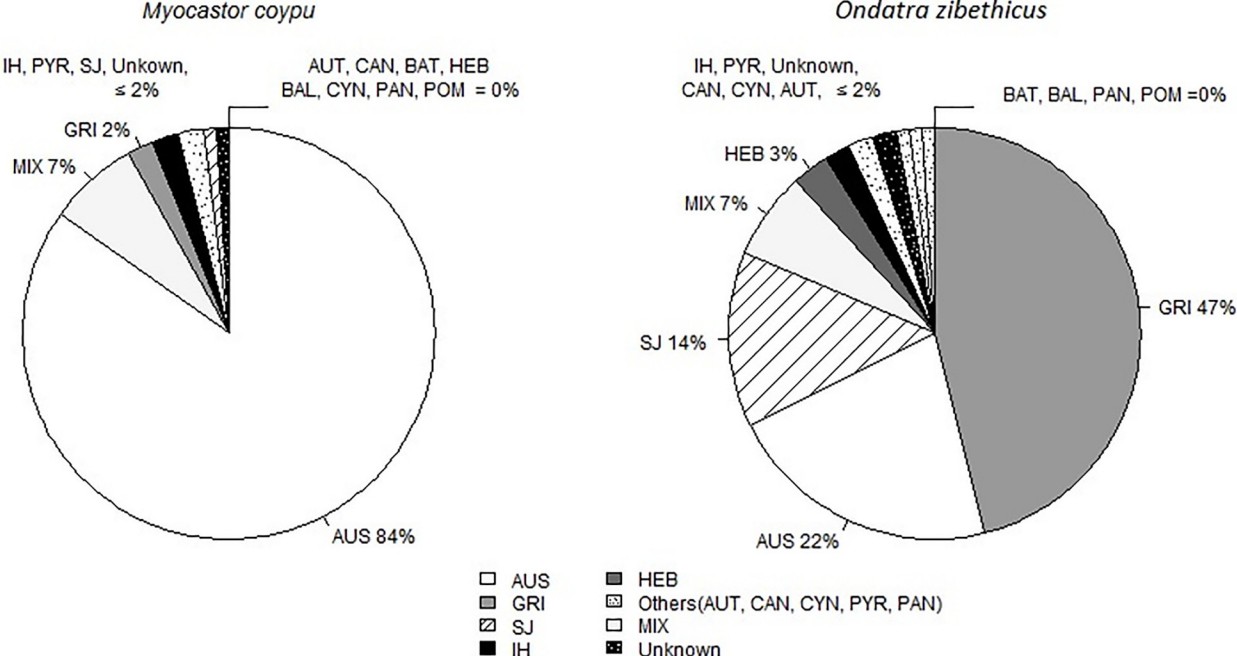

**Fig 1. Distribution of *Leptospira* serogroups among the 590 coypus and 305 muskrats tested.** Serogroup abbreviations: Australis (AUS), Autumnalis (AUT), Bataviae (BAT), Grippotyphosa (GRI), Icterohaemorrhagiae (IH), Panama (PAN), Pomona (POM), Pyrogenes (PYR), and Sejroe (SJ). MIX: results including maximum titers directed against two serogroups. Unknown: results including maximum titers directed against more than two serogroups.

combinations of Australis and Icterohaemorrhagiae (n = 4) and of Australis and Sejroe (n = 3). Among the infected muskrats (PCR-positive) with a seroreaction (n = 65), the predominant serogroup remained Grippotyphosa (34%, mixed results not included).

The spatial distribution of the predominant serogroups, Australis in coypus and Grippotyphosa in muskrats, appeared homogeneous in all three regions (Fig 2).

## Discussion

In the present study, the prevalence of renal infection by pathogenic *Leptospira* was estimated, and the distribution of serogroups was examined in the most abundant semi-aquatic rodents in Western France, coypus and muskrats. The highest prevalence was observed in muskrats and the serogroups Australis and Grippotyphosa were found to be predominant in coypus and muskrats, respectively. Considering that the field conditions can lead to PCR inhibitors in renal tissue, and infected animals may have MAT titers below the widely accepted minimum significant titer of 100 [20], prevalence and seroprevalence may have been underestimated here.

Rats are reported to be the main *Leptospira* carrier worldwide and, in France, *Leptospira* prevalence in these hosts was previously estimated to be 26% ($CI_{95\%}$: 20%-33%) [14]. Here, the results showed that the extent of *Leptospira* carriage is similar in muskrat and rat populations. Additionally, a recent outbreak of leptospirosis in Germany was linked to infection in muskrats [16], this species appearing to be a source of leptospirosis in humans. Importantly, the prevalence observed in muskrats in Western France is greater than that recently reported in Germany, varying from 3% to 13% [16].

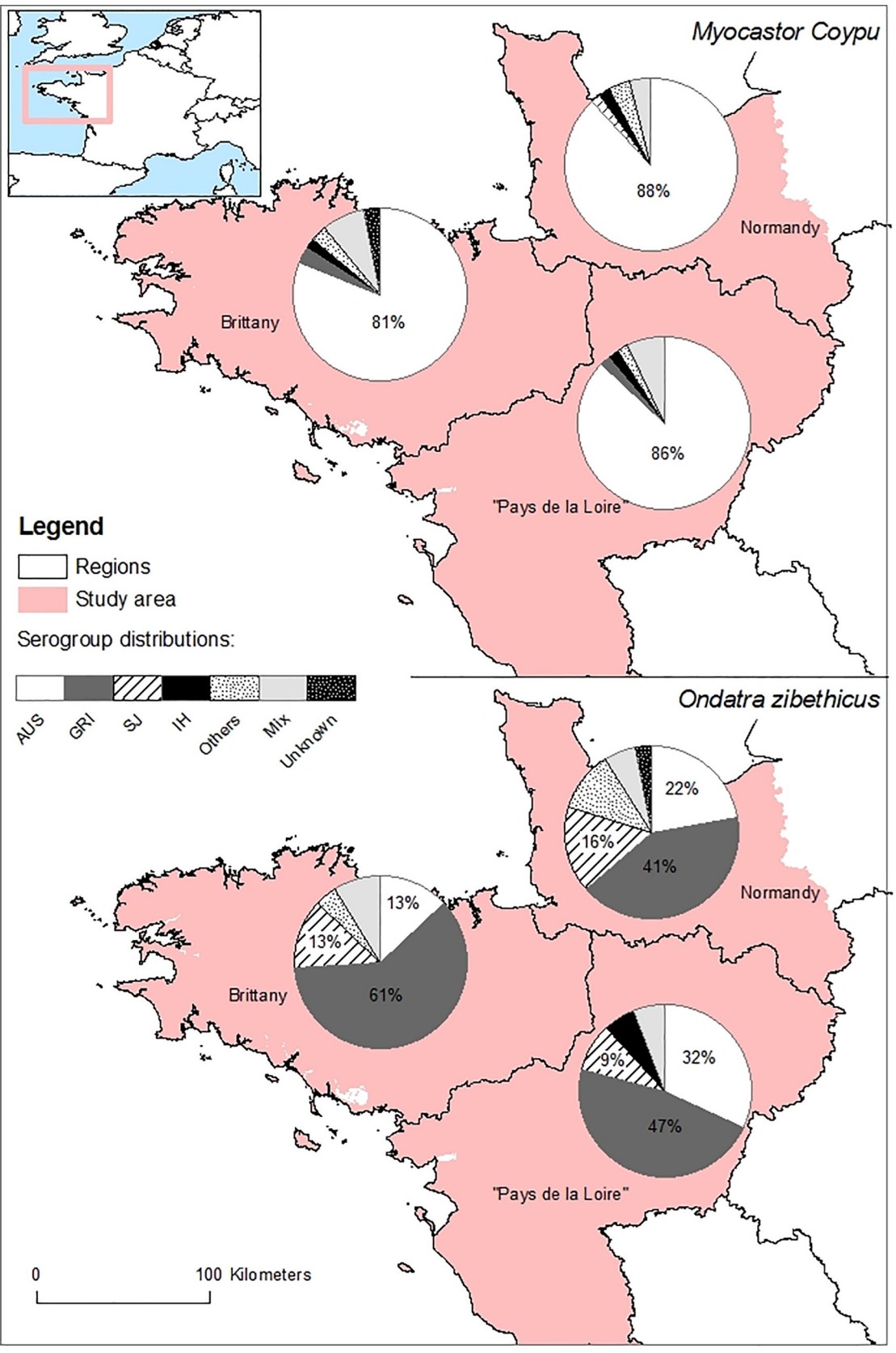

**Fig 2. Spatial distribution of the infecting serogroups.** Results obtained in coypus (n = 590) and muskrats (n = 305) in the three regions of Western France (Brittany, Normandy and Pays de la Loire). Source background map: IGN GEOFLA®.

As previously reported, the MAT correctly predicted the infecting serogroup in 46–86% of human cases [19,21]; the presumptive serogroup data appear to provide a broad overview of the serogroups commonly present in a population. Here, almost half of the muskrats tested (47%) exhibit Grippotyphosa exposure, irrespective of the region of Western France. This result is consistent with findings of a previous study in Belgium [12], giving further evidence of Grippotyphosa carriage even at a significant distance from the locations studied here. Based on the high renal carriage and the pathogenic strain of *Leptospira* found in muskrats, which was the same strain found in patients kayaking in the same region some years later [11], but also in a number of leptospirosis cases in humans (3% to 24%) and cattle (17%) in France [22,23], the presence of muskrats appears to be a considerable risk factor for humans and domestic animals.

The results in coypus are consistent with the findings of a previous study in Eastern France [15]. In this study, the same analytical methodology was used and prevalence estimates were similar to our results (12%), suggesting renal carriage by coypus in various parts of France. In addition, the majority of the coypus (84%) in the present study exhibited seroreactions to the Australis serogroup, and the consistency of distribution in different locations provided substantial evidence for Australis predominance in coypus. Serological profiles defined in a previous study in France suggested the predominance of Icterohaemorrhagiae, which raises the question of a possible switch in the serogroups mainly carried in coypus over time [24]. In France, the hedgehog has been identified as a major carrier of *Leptospira* related to the serogroup Australis and further investigations, including molecular analysis in coypus, could confirm the relative extent of Australis carriage in both species, coypu and hedgehog [25]. Antibodies against the *Leptospira* serogroup Australis, historically considered uncommon, have recently been found in 6% to 18% of infected patients, and 43% of leptospirosis cases in livestock, diagnosed in both by the use of MAT [22,23]. The present results underline the potential infectious risk for people when frequenting waters where coypus are established.

Mapping of the results suggests that the spatial distribution of the predominant serogroups, Australis in coypus and Grippotyphosa in muskrats, is homogeneous in all three regions, although both species share the same habitat. This provides evidence that these species are infected by strains of different *Leptospira* serogroups, despite exposure in the same environment. This finding is consistent with the host specificity previously described in rats [26], and suggests the ability of coypus and muskrats to carry specific *Leptospira* serogroups rather than others.

## Conclusion

This study shows that muskrats and coypus are important carriers of pathogenic *Leptospira* in aquatic environments in temperate climates. The serogroups Australis and Grippotyphosa were found to be predominant in coypus and muskrats, respectively, and the highest prevalence was observed in muskrats. Like coypus, muskrats are an invasive species of semi-aquatic rodents and their abundance can be high in water bodies possibly frequented by people and domestic animals. The presence of this species should be considered a risk factor for human and domestic animal leptospirosis and taken into account by public health policy makers, especially in terms of prevention and population control.

## Supporting information

**S1 Data. Individual information and test results.**
(XLS)

## Acknowledgments

We thank Denis Onfroy, Marie-Lucie Tropres, Gérald Guédon and Marc Pondaven from the French national federation for pest control (*Fédération national de lutte contre les organismes nuisibles*, FNLON) for the coordination of the sampling. We thank the technicians and directors of the 12 departmental federations for pest control (FDGDON) for rodent trapping, blood sampling and data collection, and the 12 departmental veterinary laboratories for sera and organ samples. We thank Jean-Marc Boucher (ANSES) for registration of the samples, as well as Claire Renaud and Océane Romatif for their contribution to sample analysis. We also thank Craig Stevens, MA, ELS for English editing.

## Author Contributions

**Conceptualization:** Angeli Kodjo, Franck Boué, Céline Richomme.

**Formal analysis:** Florence Ayral.

**Funding acquisition:** Franck Boué.

**Investigation:** Gérald Guédon.

**Methodology:** Franck Boué, Céline Richomme.

**Resources:** Gérald Guédon.

**Software:** Angeli Kodjo.

**Supervision:** Céline Richomme.

**Visualization:** Florence Ayral.

**Writing – original draft:** Florence Ayral, Céline Richomme.

**Writing – review & editing:** Florence Ayral, Angeli Kodjo, Gérald Guédon, Franck Boué, Céline Richomme.

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
