## [Decision Letter · Decision Letter 0]

6 Nov 2019

PONE-D-19-24496

Muskrats are greater carriers of pathogenic Leptospira than coypus in ecosystems with temperate climates

PLOS ONE

Dear Dr. Ayral,

Thank you for submitting your manuscript to PLOS ONE. After careful consideration, we feel that it has merit but does not fully meet PLOS ONE’s publication criteria as it currently stands. Therefore, we invite you to submit a revised version of the manuscript that addresses the points raised during the review process.

ACADEMIC EDITOR: 

Appropriate ethical clearance approval details must be given in the methods for the study conductance. The experimental result validation by proper statistics is needed to validate the results. 

We would appreciate receiving your revised manuscript by December 10, 2019. To enhance the reproducibility of your results, we recommend that if applicable you deposit your laboratory protocols in protocols.io, where a protocol can be assigned its own identifier (DOI) such that it can be cited independently in the future. For instructions see: http://journals.plos.org/plosone/s/submission-guidelines#loc-laboratory-protocols

We look forward to receiving your revised manuscript.

Kind regards,

Kalimuthusamy Natarajaseenivasan

Academic Editor

PLOS ONE

Journal Requirements:

1. We note that [Figure(s) 2] in your submission contain [map/satellite] images which may be copyrighted. All PLOS content is published under the Creative Commons Attribution License (CC BY 4.0), which means that the manuscript, images, and Supporting Information files will be freely available online, and any third party is permitted to access, download, copy, distribute, and use these materials in any way, even commercially, with proper attribution. For these reasons, we cannot publish previously copyrighted maps or satellite images created using proprietary data, such as Google software (Google Maps, Street View, and Earth). For more information, see our copyright guidelines: http://journals.plos.org/plosone/s/licenses-and-copyright.

1.    You may seek permission from the original copyright holder of Figure(s) [2] to publish the content specifically under the CC BY 4.0 license. 

Reviewers' comments:

Reviewer's Responses to Questions

**Comments to the Author**

1. Is the manuscript technically sound, and do the data support the conclusions?

Reviewer #1: Yes

Reviewer #2: Yes

2. Has the statistical analysis been performed appropriately and rigorously? 

Reviewer #1: N/A

Reviewer #2: Yes

3. Have the authors made all data underlying the findings in their manuscript fully available?

Reviewer #1: Yes

Reviewer #2: Yes

4. Is the manuscript presented in an intelligible fashion and written in standard English?

Reviewer #1: No

Reviewer #2: No

5. Review Comments to the Author

Reviewer #1: The manuscripts is accepted for publication but correction must be made prior to being accepted for publication.

The manuscript is well written but the author used "we" in the written manuscript and that should be avoided. Other words that can be use such as "This study, Present study, In this study and etc" would be more approriate.

Grammars still need to be looked through properly. The word individual that refers to rodent were inappropriate.

The conclusion and the statistical analysis is missing.

Reviewer #2: author can be obtain a bio safety ethics committee permission for conduction of trial ?

6. PLOS authors have the option to publish the peer review history of their article (what does this mean?). If published, this will include your full peer review and any attached files.

Reviewer #1: No

Reviewer #2: No

---

## [Author Response · Author response to Decision Letter 0]

13 Dec 2019

Marcy L’Étoile, Decembre 6, 2019

Dear Editor,

As suggested, we ensured that our manuscript met PLOS ONE's style requirements. We thus used Level 1 headings for all major sections (Abstract, Introduction, Materials and Methods, Results, Discussion, etc.), bold type, 18pt font. We modified the Figure citations and the figure captions according to the guidelines.

Considering your comment on [Figure(s) 2] that contains [map/satellite] images, which may be copyrighted, we fully understand the issue related to copyrights. The mapping in Figure 2 was generated using ArcGIS® and the background map: IGN GEOFLA®. We therefore cited the source as follows:

“To assess potential variation in Leptospira serogroup distribution, the study area was divided into three administrative regions: Brittany, Normandy and Pays de la Loire. Data were visualized in ArcGIS version 9.3 (ESRI, Redland, CA, USA) with the background map from IGN GEOFLA®.”

The figure caption also includes “Source background map: IGN GEOFLA®.”

Please find below the point-by-point responses to each of the three reviewers’ comments.

Reviewers #1 and #2: 

Is the manuscript presented in an intelligible fashion and written in standard English? Answers were “No”

To improve the flow of the paper, the phrasing and grammar, we have asked a native English speaker, Editor in the Life Sciences, to revise our manuscript. Required changes to grammar and style were made.

Reviewers #1: 

Comments 1, 2, 4, 5, 7, 8, 9, 13 and 15 were related to typing errors or inappropriate phrasing. For each of his comment, the reviewer kindly gave some suggestions.

We agree with these remarks and modified the manuscript accordingly. We thank the reviewer for these suggestions which increase general understanding of the paper.

Comment 3: “Author mentioned that “No animal” were killed. What does that mean? How was the kidney isolated for the study then? That is worrisome.”

We agree that the ethics statement must be clarified to ensure clear understanding of the study context. We thus amended the sentence as follows:

“All samples were collected from rodents legally killed for population control; therefore, this study did not involve deliberate additional killing of animals and no ethical approval was considered necessary. All procedures for population control complied with the ethical standards of the relevant national and European regulations on the care and use of animals (French authority Decision 2007/04/06 and Directive 2010/63/EC).”

Comment 6: “There were no information on how and what data was use for analysis. In the result section, there were percentage and confident interval used.”

We agree with the reviewer’s comment. The information on how we obtained the prevalence, the seroprevalence and the confident intervals was missing; we therefore included the following sentence in the Materials and Methods section:

“Apparent prevalence, seroprevalence, and 95% confidence intervals were calculated using exact binomial tests.”

Comment 10: “Wondering if the word infection in muskrat and infection in rat was appropriate. These animals are clinically healthy right? Rodents are known to be carrier and they just transmit the disease. “

We thank the reviewer who has underlined that using ‘infection’ in rodents could be confusing or inappropriate. To clarify this point, we amended the sentence as follows: 

“Here, the results showed that the extent of Leptospira carriage is similar in muskrat and rat populations.”

Comment 11: « line 134. “cases” These cases refer to human cases or animal cases.”

We agree that the “cases” needed to be clarified. We thus added “human cases” as the specificity stated was related to analysis in humans.

Comment 12:”line 141. All the percentages. What do they mean. Which one is for human and which percentage is for the livestock? Which type of livestock? Need clarity or detail.”

We agree that this sentence should be clarified to improve the flow of the paper and for a better understanding of the reader. We amended the sentence as follows:,

“Based on the high renal carriage and the pathogenic strain of Leptospira found in muskrats, which was the same strain found in patients kayaking in the same region some years later, but also in a number of leptospirosis cases in humans (3% to 24%) and cattle (17%) in France...”

Comment 14: “line 155. The present study confirms…How did this study confirm the infection risk. There were no human being samples. You are merely referring to the other study and both studies had occurred on two different time point.”

We agree that “confirm” was not appropriate. We modified the sentence as follows:

“[…] the presence of muskrats appears to be a considerable risk factor for humans and domestic animals.”

Comment 16 and 17: One sentence as a stand-alone paragraph. Please check.

Where is the conclusion?

We agree that a conclusion could improve the flow of the paper and we followed the reviewer suggestion. We thus included the following conclusion,

“This study shows that muskrats and coypus are important carriers of pathogenic Leptospira in aquatic environments in temperate climates. The serogroups Australis and Grippotyphosa were found to be predominant in coypus and muskrats, respectively, and the highest prevalence was observed in muskrats. Like coypus, muskrats are an invasive species of semi-aquatic rodents and their abundance can be high in water bodies possibly frequented by people and domestic animals. The presence of this species should be considered a risk factor for human and domestic animal leptospirosis and taken into account by public health policy makers, especially in terms of prevention and population control.”

---

## [Decision Letter · Decision Letter 1]

21 Jan 2020

Muskrats are greater carriers of pathogenic Leptospira than coypus in ecosystems with temperate climates

PONE-D-19-24496R1

Dear Dr. Ayral,

We are pleased to inform you that your manuscript has been judged scientifically suitable for publication and will be formally accepted for publication once it complies with all outstanding technical requirements.

With kind regards,

Kalimuthusamy Natarajaseenivasan

Academic Editor

PLOS ONE

Additional Editor Comments (optional):

Reviewers' comments:

Reviewer's Responses to Questions

**Comments to the Author**

1. If the authors have adequately addressed your comments raised in a previous round of review and you feel that this manuscript is now acceptable for publication, you may indicate that here to bypass the “Comments to the Author” section, enter your conflict of interest statement in the “Confidential to Editor” section, and submit your "Accept" recommendation.

Reviewer #1: All comments have been addressed

Reviewer #2: All comments have been addressed

2. Is the manuscript technically sound, and do the data support the conclusions?

Reviewer #1: Yes

Reviewer #2: Yes

3. Has the statistical analysis been performed appropriately and rigorously? 

Reviewer #1: Yes

Reviewer #2: N/A

4. Have the authors made all data underlying the findings in their manuscript fully available?

Reviewer #1: Yes

Reviewer #2: Yes

5. Is the manuscript presented in an intelligible fashion and written in standard English?

Reviewer #1: Yes

Reviewer #2: Yes

6. Review Comments to the Author

Reviewer #1: Changes has been made based on the comments given. This article has been improved. The language "English" is now much better. The statistical analysis in the material and methods and results has been explained with better clarity.

Reviewer #2: (No Response)

7. PLOS authors have the option to publish the peer review history of their article (what does this mean?). If published, this will include your full peer review and any attached files.

Reviewer #1: No

Reviewer #2: Yes: Dattatarya Kadam

---

## [Editor Report · Acceptance letter]

29 Jan 2020

PONE-D-19-24496R1 

Muskrats are greater carriers of pathogenic *Leptospira* than coypus in ecosystems with temperate climates 

Dear Dr. Ayral:

I am pleased to inform you that your manuscript has been deemed suitable for publication in PLOS ONE. Congratulations! Your manuscript is now with our production department. 

With kind regards,

on behalf of

Dr. Kalimuthusamy Natarajaseenivasan 

Academic Editor

PLOS ONE